# Prevention Effect of TGF-β Type I Receptor Kinase Inhibitor in Esophageal Stricture Formation after Corrosive Burn

**Min-Tae Kim** [1,2] and **Kun-Yung Kim** [1,3,*]

1 Biomedical Engineering Research Center, Asan Medical Center, University of Ulsan College of Medicine, 88, Olymic-Ro 43-Gil, Songpa-gu, Seoul 05505, Korea; noir09@gmail.com
2 Department of Radiologic Technology, Cheju Halla University, Jeju-si 63092, Korea
3 Department of Radiology, Chonbuk National University Hospital, Research Institute of Clinical Medicine, Biomedical Research Institute of Chonbuk National University Hospital, 20, Geonji-ro, Deokjin-gu, Jeonju-si 54907, Korea
* Correspondence: kky2kkw@jbnu.ac.kr; Tel.: +82-10-5063-0046

**Abstract:** Corrosive burns lead to progressive esophageal stricture and dysphagia. There are many trials to prevent esophageal stricture formation after corrosive burn. EW-7197 has been proven in several animal models of fibrosis to have antifibrotic and antiproliferative effect. This study aimed to assess the effects of EW-7197 on prevention for esophageal stricture formation after corrosive esophageal burn. An animal study was carried out, where the animals were divided into three groups: a healthy group, a control group (corrosive burn without EW-7197), and a treatment group (corrosive burn with EW-7197). Corrosive esophageal burns were induced using 30% NaOH on the lower esophagus. For 3 weeks, the control group received vehicle and the treatment group received 20 mg/kg/day EW-7197. Treatment efficacy was assessed by measuring the stenosis ratio by esophagogram with contrast media on day 21. Histologic staining was performed to evaluate the fibrosis area ratio, and Western blotting was performed to evaluate fibrotic markers. Among 20 rats that underwent surgery, 14 survived. Three in the treatment group died because of esophageal perforation, and three in the control group died due to their debilitating status. The esophageal stenosis ratio was significantly lower in the treatment group than in the control group ($12.1 \pm 9.5\%$ and $42.2 \pm 8.3\%$, respectively; $p = 0.001$). The histologic fibrosis area ratio was also significantly lower in the treatment group ($12.5 \pm 3.0\%$ and $21.6 \pm 2.1\%$, respectively; $p = 0.001$). The treatment group showed lower expressions of profibrogenic proteins such as TGF-β1, pSmad3, and α-SMA. EW-7197 may be a good alternative for the prevention esophageal stricture formation after corrosive burn.

**Keywords:** corrosive burn; esophageal stricture; TGF-β1

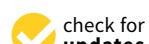



## 1. Introduction

Ingestion of corrosive substances accidentally or in attempted suicide remains a significant cause of esophageal strictures [1,2]. Esophageal stricture formation after corrosive burn is a complication of deep esophageal injury, which stimulates excess fibrogenic actions. This leads to progressive esophageal stricture and dysphagia, which significantly decreases quality of life. A number of therapeutic options have been investigated to prevent the formation of strictures in experimental corrosive esophageal injury models [3–9]. The majority of these therapeutic agents have been used to exert antifibrogenic actions, but the benefits and standard protocols are still controversial [4]. In addition, most of these options have not obtained clearance for clinical use because of suspected toxicities, such as immunosuppression.

The transforming growth factor (TGF)-β type I receptor kinase inhibitor family has been tested in various animal models for fibrosis [10–14]. EW-7197 (N-[[4-([1,2,4] Triazolo[1,5-a]pyridin-6-yl)-5-(6-methylpyridin-2-yl)-1H-imidazol-2-yl]methyl]-2-fluoroaniline) is a novel orally available TGF-β type I receptor kinase inhibitor with high selectivity and low toxicity.

EW-7197 has been investigated in several animal models of fibrosis to have antifibrotic and antiproliferative effect [15,16].

We hypothesized that EW-7197 can suppress stricture formation after corrosive esophageal burn by inhibiting TGF-β type I-induced deposition of the extracellular matrix (ECM). Therefore, this study aimed to access the effects of EW-7197 on prevention for esophageal stricture formation after corrosive burn.

## 2. Materials and Methods

### 2.1. Antifibrotic Drug

EW-7197 phosphate was provided from the Laboratory of Medicinal Chemistry, College of Pharmacy, Ewha Woman's University (Seoul, Korea). The dosage of EW-7197 phosphate was decided, based on previous studies [17–19].

### 2.2. Animals

All animal experiments were approved by the Institutional Animal Care and Use Committee at the Asan Medical Center, University of Ulsan College of Medicine (No. 2017-13-246). A total of 30 Sprague Dawley rats (9 weeks old, 300–350 g) were divided using a random allocation program (Microsoft, Seattle, WA, USA) into 3 groups: a healthy group, a corrosive burn without EW-7197 treatment group (the control group), and a corrosive burn with EW-7197 treatment group (the treatment group).

### 2.3. Surgical Procedure

Surgical procedure was performed as previously described [5]. After 12 h of fasting, rats were anesthetized by intramuscular injection of 50 mg/kg zolazepam and tiletamine (Zoletil 50; Virbac, Carros, France) and 10 mg/kg xylazine (Rompun; Bayer HealthCare, Leverkusen, Germany). A median laparotomy was performed, and a 1.5 cm segment of the lower esophagus was prepared after dissection. A 5 French feeding catheter was placed in the upper part of the lower esophagus through the mouth. The gastroesophageal junction was tied externally with 2/0 silk ligature to prevent leakage of caustic agent into the stomach. Proximally, just under the diaphragm, the esophagus was tied externally with 2/0 silk ligature to prevent regurgitation and aspiration. A thinner catheter was inserted through the feeding catheter for irrigation. Then, 1 mL of 30% NaOH solution was injected through the thinner catheter for 90 s. Subsequently, the solution was aspirated back, and distilled water was used to irrigate the burned area for 60 s. Intraluminal pressure were carefully controlled during infusion and perfusion until slight translucency of the esophageal wall and branching of vessels were observed. The proximal silk ligature was cut, and the catheter was pulled out under negative pressure. The distal suture was then cut, and the laparotomy was finished. All animals were kept on a standard rodent pellet diet with tap water ad libitum after surgery. The treatment group (n = 10) received 20 mg/kg EW-7197 phosphate dissolved in 0.3 mL vehicle by gavage once daily for 3 weeks after surgery. The control group (n = 10) received a solution consisting of 0.3 mL of artificial gastric juice once daily for 3 weeks after surgery. The healthy group (n = 10) was not subjected to the surgical procedure and did not receive any treatment. After surgical procedure, behavioral and weight changes were monitored weekly. At 3 weeks after surgical procedure, all rats sacrificed for histopathology and Western blotting by administering inhalable pure carbon dioxide.

### 2.4. Fluoroscopic Esophagogram

Fluoroscopic esophagogram with contrast media was performed at 3 weeks after the procedure in all groups. Esophagogram was performed under the same anesthesia method as described above, using a contrast medium (Ultravist 300; Schering Korea, Anseong, Korea). The stenosis ratio was evaluated by measuring the axial diameter in both the upper non-damaged area and the lower stenosis area. Quantitative determination was performed using Image J software.

### 2.5. Esophageal Histopathology

The stenosed portion of the lower esophagus was obtained 3 weeks after corrosive burn. Tissue samples were fixed in 10% neutral-buffered formalin for 24 h, which was then embedded in paraffin, sectioned into 5-μm sections, and stained with Masson's trichrome. The healthy rats underwent the same procedure. The total areas encircling the outer margins of the esophagus, luminal area, and submucosal collagen area of the axial section were measured to determine the fibrosis area ratio. This ratio was defined by the formula (submucosal collagen area)/(total area − luminal area). All histopathologic images were analyzed with the Image J software.

### 2.6. Western Blotting

Western blotting was performed using 10% sodium dodecyl sulfate polyacrylamide gel electrophoresis. Primary antibodies were against Smad3 (1:1000; Cell Signaling Technology (CST), Danvers, MA, USA), pSmad3 (1:1000; CST), $\alpha$-smooth muscle actin ($\alpha$-SMA; 1:300; Abcam, Cambridge, Cambridgeshire, UK), TGF-$\beta$1 (1:1000; CST), and $\beta$-actin (1:1000; CST). After incubation with horseradish peroxidase-conjugated secondary antibodies (1:1000; Jackson ImmunoResearch, West Grove, PA, USA) and enhanced chemiluminescence Western blotting detection reagents (Amersham Biosciences, Little Chalfont, Buckinghamshire, UK), immunoreactive bands were visualized using an Ez-Capture MG (ATTO Corporation, Tokyo, Japan). Densitometric values of the bands were quantified and expressed as their ratio to $\beta$-actin using CS Analyzer software (ATTO Corporation). The healthy group was used to ascertain normal values.

### 2.7. Statistical Analysis

Differences between the groups were analyzed using the Kruskal–Wallis test and the Mann–Whitney U test. Body weight measurement was carried out in the rats that survived the full experimental period. A $p$ value of <0.05 was considered statistically significant. Statistical analyses were performed using the SPSS software (version 23.0; SPSS, IBM, Chicago, IL, USA).

## 3. Results

Each procedure taken approximately 30 min for completion. Among 20 rats that underwent surgery, 14 survived the complete experimental period of 3 weeks. A total of six rats died: three in the treatment group, from esophageal perforation, and three in the control group, from a debilitating status related to dysphagia, in the second week. The 21-day survival rates were 70% in both the control and treatment groups. Baseline body weight was not statistically significantly different between the control and treatment groups (343.2 ± 32.8 and 329.5 ± 22.2, respectively; $p = 0.490$), and body weight gain was not statistically significantly different between the two groups (12.6 ± 72.8 and −19.7 ± 50.8, respectively; $p = 0.384$). The prevalence rate of esophageal stricture-related symptoms (which included hypersalivation, hemoptysis, and stridor) was 50% (n = 7; 3 rats in the control group and 4 in the treatment group). The remaining seven rats were free from esophageal stricture-related symptoms.

### 3.1. Effect of the Drug on Esophageal Stenosis

Figure 1 shows the esophagographic stenosis ratio, as determined by fluoroscopic esophagogram. The stenosis ratios of the esophagus in the healthy group, control group, and treatment group were 1.3 ± 5.2%, 42.2 ± 8.3%, and 12.1 ± 9.5%, respectively. The treatment group had a significantly lower stenosis ratio than the control group ($p = 0.001$).

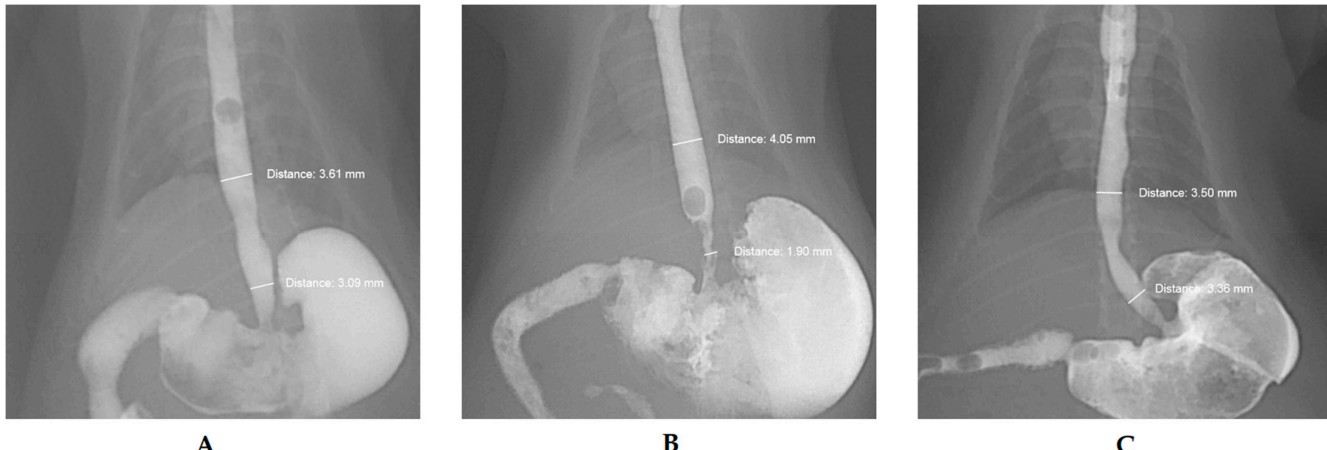

**Figure 1.** Fluoroscopic esophagogram with contrast media at 3 weeks after surgery. The stenosis ratio was determined by comparing the axial diameter in both the upper undamaged area and the lower stenosis area. Healthy rat; rat with corrosive burn but without TGF-β1 inhibitor treatment (control); rat with corrosive burn and with TGF-β1 inhibitor treatment (treatment) ((**A**–**C**), respectively). The figure demonstrates significantly less esophageal stenosis in the treatment group compared with the control group ($p = 0.001$).

### 3.2. Effect of the Drug on Fibrosis

Figure 2 shows the fibrosis area ratio, as determined by histopathologic results. The fibrosis area ratios of the esophagus in the healthy group, control group, and treatment group were $10.6 \pm 3.2\%$, $21.6 \pm 2.1\%$, and $12.5 \pm 3.0\%$, respectively. The treatment group had a significantly lower fibrosis area ratio than the control group ($p = 0.001$).

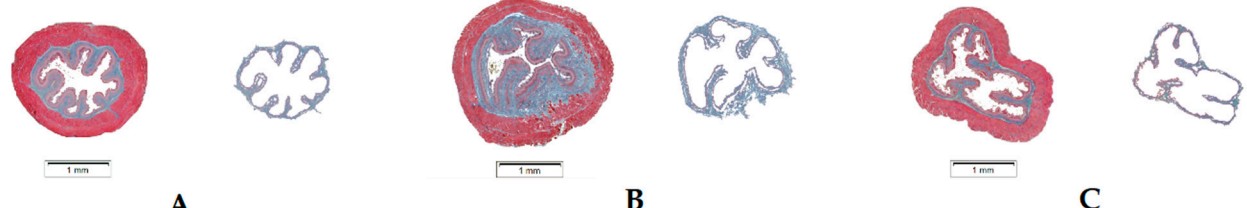

**Figure 2.** Masson's trichrome stain was used for fibrosis determination. Healthy rat; rat with corrosive burn but without TGF-β1 inhibitor treatment (control); rat with corrosive burn and with TGF-β1 inhibitor treatment (treatment) ((**A**–**C**), respectively). The fibrosis area ratio was significantly lower in the treatment group than that in the control group ($p = 0.001$).

### 3.3. Effect of the Drug on Protein Expression

Figure 3 shows the Western blotting results. EW-7197 treatment for 3 weeks significantly suppressed the expression of the profibrogenic proteins TGF-β1, pSmad3, and α-SMA. We found that the expression of these proteins increased significantly in the control group compared with the treatment group ($p < 0.05$).

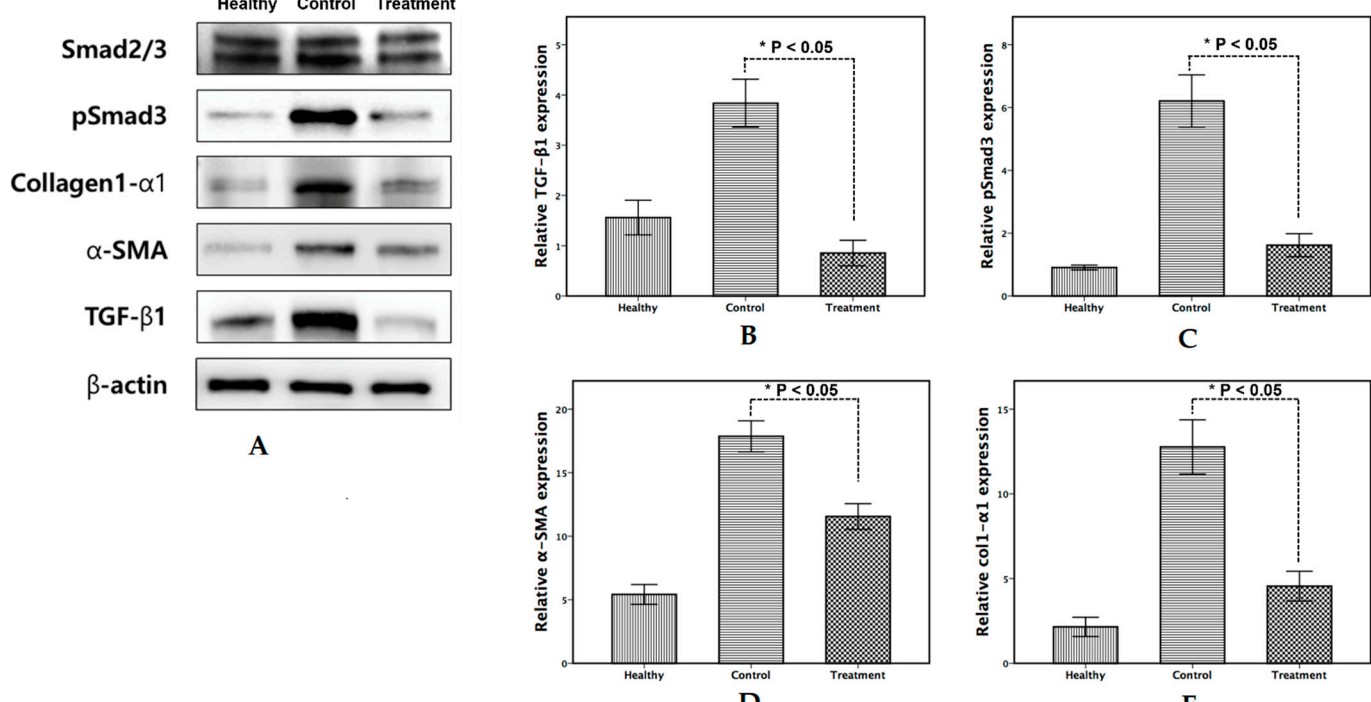

**Figure 3.** Representative Western blotting results (**A**). Quantitative expression ratio of TGF-β1, pSmad3, and α-SMA were significantly decreased in the treatment group compared with the control group ((**B**–**E**), respectively) ($p = 0.001$).

## 4. Discussion

Stricture formation in the chronic stage of corrosive esophagitis is a commonly encountered complication. It requires long and repetitive treatment. Some investigators advocate insertion of a silicone stent or a nasogastric tube for the acute and subacute stages after a corrosive burn to reduce stricture formation [20,21]. However, Gumaste VV et al. do not advocate these practices, because stent-induced mechanical injury may contribute to substantial complications [22]. Although steroids are clinically used, they expose patients to many side effects and controversial results. There is no established medical treatment during the acute stage that can prevent further stricture formation. Experimentally, many treatment options have been used to reduce stricture formation [3–9,23]; however, only a few of them are used clinically.

Antifibrotic agents have been the main focus of preventing esophageal stricture. Recently, Orozco-Perez et al. reported that pirfenidone, an ECM degradation modulator, prevented esophageal stricture formation after corrosive burn in rats [9]. The collagen deposition in the tissue is controlled between the synthesis and decomposition of various types of collagen components of the ECM [19]. Collagen synthesis is enhanced by TGF-β1. It plays an important role in the transition from inflammatory phase to proliferative phase, particularly in fibroblasts and collagen deposition, indicating the pathophysiological pathway of stenosis formation [23].

Kim et al. reported that blocking the biological action of TGF-β may be by an alternative treatment option to inhibit significant stricture formation [14,24,25]. More recently, Jun et al. reported that EW-7197 suppressed fibrotic tissue hyperplasia formation with esophageal stent in a rat esophageal model [19]. They suggested it has potential treatment option as an antifibrotic drug via its ability to inhibit TGF-β signaling [19]. EW-7197 is a selective oral bioavailable inhibitor of TGF-β1, which has been studied as an antifibrotic agent [15,16,18]. In this context, we hypothesized that a TGF-β1 inhibitor could suppress stricture formation after corrosive esophageal burn by inhibiting TGF-β1/Smad3-induced deposition of the ECM. The pharmacologic effects of EW-7197 could allow its application to corrosive strictures when all other treatment options are not adequate.

Our study demonstrated that the stenosis ratio determined by the fluoroscopic study and the fibrosis area ratio determined by histology analysis were significantly lower in the treatment group than those in the control group. Although we did not perform quantitative analysis, the expression patterns of profibrogenic proteins such as pSmad3 and α-SMA were suppressed in the treatment group in our Western blot analysis. This result indicates that EW-7197 suppressed stricture formation by inhibiting TGF-β1/Smad3-induced ECM synthesis. Our results are consistent with previous studies performed in various animal models [14–16,18].

However, our results were not directly connected to the improvement of survival or symptoms. Survival rates at 3 weeks after surgery were the same (70%) in both groups. In addition, the occurrence rate of esophageal stricture-related symptoms was 42.9% in the control group and 57.1% in the treatment group. Moreover, three rats in the treatment group died in the first week after surgery due to esophageal perforation. TGF-β could be responsible, not only for wound scarring but also for wound healing. The key roles of TGF-β in wound healing including production and remodeling of the ECM, which is crucial for tissue damage repair. We surmise that the treatment group experienced delayed wound healing and fibrosis inhibition in the injured segment, which led to esophageal perforation. It is already well known that fibroblast proliferation and collagen deposition occur in the second week after injury, representing the biological pathway of stricture formation [25]. More research is needed to investigate complex etiological causes in acute and subacute periods of corrosive burns.

Our study included some limitations. First, we excluded three rats that died due to their debilitating status because they did not survive until their scheduled sacrifice date. However, because these rats may have represented the most severe corrosive injury cases, this could have biased the results. Nevertheless, we believe these deaths were consequences of severe injury, rather than death related to stricture formation, because the timing of the deaths was concentrated in the first week after the corrosive burn. Second, it is possible that peritoneal adhesion formation after surgery may have affected the fluoroscopic results. We used a modified version of Gehanno and Guedon's experimental model, the most commonly used model in previous studies [26]. In this model, the lower esophagus is dissected and ligated to prevent leakage of the caustic substance. Adhesion between the serosa of the esophagus and peritoneum occurred during the healing period. To minimize this adhesion, we meticulously dissected the overlying peritoneum to avoid adjacent tissue damage. Finally, we did not observe interobserver variability regarding the analyses of fluoroscopic and histologic findings.

In conclusion, EW-7197 may be a good alternative treatment option for the prevention of esophageal stricture formation after corrosive burn. Although our results are prospecting, subsequent research is needed to investigate the safety of this TGF-β1 inhibitor and to establish an optimized treatment protocol in preventing esophageal strictures after corrosive burns.

**Author Contributions:** Conceptualization, M.-T.K. and K.-Y.K.; methodology, M.-T.K.; software, K.-Y.K.; validation, M.-T.K. and K.-Y.K.; formal analysis, M.-T.K.; investigation, M.-T.K.; resources, K.-Y.K.; data curation, K.-Y.K.; writing—original draft preparation, M.-T.K.; writing—review and editing, K.-Y.K.; visualization, M.-T.K.; supervision, K.-Y.K.; project administration, K.-Y.K.; funding acquisition, M.-T.K. and K.-Y.K. All authors have read and agreed to the published version of the manuscript.

**Funding:** This research was supported by the National Research Foundation of Korea (NRF) grant funded by the Korea government (MSIP; Ministry of Science, ICT & Future Planning) (NRF-2018R1D1A1A02043422), (NRF-2019R1F1A1040357) and was supported by Fund of Biomedical Research Institute (CUH2020-0010), Chonbuk National University Hospital.

**Institutional Review Board Statement:** All animal experiments were approved by the Institutional Animal Care and Use Committee at the Asan Medical Center, University of Ulsan College of Medicine (No. 2017-13-246).

**Informed Consent Statement:** No applicable.

**Conflicts of Interest:** The authors declare no conflict of interest.

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
