# Peer review of "Prevention Effect of TGF-β Type I Receptor Kinase Inhibitor in Esophageal Stricture Formation after Corrosive Burn"

_applsci, doi:10.3390/app112311536_

Round 1

Reviewer 1 Report

It is interesting findings. I have only have minor comments:

  • Some readers might be clinician, please elaborate more the clinical application of findings in the Discussion section.
  • Methods: please clarify whether the study met the animal welfare guidelines.

Author Response

Reviewer #1: R 1-1. Some readers might be clinician, please elaborate more the clinical application of findings in the Discussion section.

Response: These have been added, as suggested.

R 1-2. please clarify whether the study met the animal welfare guidelines.

Response: The study was conducted in compliance with the Animal Welfare Guidelines.

Reviewer 2 Report

Study, concept,method, drug and figures are well  presented.

Author Response

Thank you for referees approval

Reviewer 3 Report

The authors have carried out a convincing study on the use of a TGF-β Type 1 Receptor Kinase inhibitor drug in inhibiting esophageal stricture formation using a rodent model. They use the drug EW-7197 on rats where the stenosis was induced using NaOH and examine for the reduction in stricture formation, when compared to controls. To further validate that the reduction in strictures was through differential expression of profibrogenic proteins, the authors have probed for the expression levels of pSmad and α-SMA and suggested a mechanism of action for EW-7197.

Major points:

Abstract

At the start of the abstract, a small justification for examining the effect of the drug EW-1797 would be good.

Line 14: Should be ‘An animal study was done where the animals were divided into three groups’. They don’t mention the animal used. Leaves the reader confused.

Introduction

The authors have mentioned the basis of focusing on the TGF-β Type 1 receptor kinase in it’s’ role in fibrosis as a pathway to inhibit in the quest for treatment for esophageal strictures. A little more information on the biology of the pathway as a potential therapeutic target would be relevant. Although the authors mention EW-7197 as the drug in focus and reference it in its development and use, some literature on the action of the drug would be good.

Materials and methods

Section 2.2 Animals (line 59): mentions that thirty rats were used. The abstract mentions 20 rats being used. This is contradictory.

Section 2.3 Surgical procedure (Line 81): They mention the drug being administered at a 20 mg/Kg dosage. The control group was administered with artificial gastric juice. Was the artificial gastric juice used as a vehicle? If so, was the drug administered to the treatment group dissolved in the vehicle?

The authors mention that they decided on the dosage to use based on previous studies. These studies have been done on other organs in the context of other diseases. Did the authors try a dose response curve to optimize the dosage to use for the current study?

Also, how did they decide on the three week time period of drug treatment, after which to test for the effect of the drug?  Did they consider the time period for strictures to start forming?

Results

Line 127: Sentence should be ‘baseline body weight was not statistically significantly different between the control and treatment groups and body weight gain was not statistically significantly different between the two groups.’ – Minor comments

The effect of the drug on preventing corrosive strictures seems convincing. It would be know at what point of the three week period the corrosion starts and the drug starts to take effect, if possible.

Figure 3:

Fig 3-A does not indicate what the individual columns represent.

Fig 3B- 3E: Statistically significant groups should be indicated using asterisks. For example are the ‘Healthy’ and ‘Treatment’ groups not significantly different in all the histograms? There seems to be a significant difference between the groups in Figures 3D and 3E.

Discussion

Line 192: ‘were prominent lower in the 192 treatment group than those in the control group’ may need to be rephrased to clarify.

Minor points:

Line 32: ‘Formation’ should be ‘formation’ – Minor comments

Section 2.3. Surgical procedure (Line 70): repeat phrase ‘to prevent’ should be removed. – minor comments

Section 2.3. Surgical procedure (Line 85): Sentence should be ‘behavioral and weight changes were monitored weekly.’ – Minor comments

Line 182: ‘Kim et al. reported that blocking the biological action of TGF-β man by an alternative’

Line 206: ‘We surmise that the treatment group experienced (with? Delete word) delayed would healing…’

Line 212: ‘Our study included some limitations’

Author Response

Reviewer #3: R 3-1. At the start of the abstract, a small justification for examining the effect of the drug EW-1797 would be good.

Response: These have been added, as suggested.

R 3-2. Line 14: Should be ‘An animal study was done where the animals were divided into three groups’. They don’t mention the animal used. Leaves the reader confused.

Response: These have been added and revised, as suggested.

R 3-3. The authors have mentioned the basis of focusing on the TGF-β Type 1 receptor kinase in it’s’ role in fibrosis as a pathway to inhibit in the quest for treatment for esophageal strictures. A little more information on the biology of the pathway as a potential therapeutic target would be relevant. Although the authors mention EW-7197 as the drug in focus and reference it in its development and use, some literature on the action of the drug would be good.

Response: In our opinion, reference regarding this potential therapeutic target is sufficient.

R 3-4. Section 2.2 Animals (line 59): mentions that thirty rats were used. The abstract mentions 20 rats being used. This is contradictory.

Response: In abstract, we comment animal study into three groups. Healthy group, Control group, Treatment group. Only 20 rats receive surgery with corrosive burn except Healthy group.

R 3-5. Section 2.3 Surgical procedure (Line 81): They mention the drug being administered at a 20 mg/Kg dosage. The control group was administered with artificial gastric juice. Was the artificial gastric juice used as a vehicle? If so, was the drug administered to the treatment group dissolved in the vehicle?

Response: Artificial gastric juice used as a vehicle. EW-7197 phosphate dissolved in 0.3 ml vehicle by oral gavage. These have been added, as suggested.

R 3-6. The authors mention that they decided on the dosage to use based on previous studies. These studies have been done on other organs in the context of other diseases. Did the authors try a dose response curve to optimize the dosage to use for the current study?

Response: These have been added related similar organ study reference, as suggested.

R 3-7. Also, how did they decide on the three week time period of drug treatment, after which to test for the effect of the drug?  Did they consider the time period for strictures to start forming?

Response: On this comment, TGF-b1 is a member of the TGF-b superfamily that controls processes such as proliferation. Our research is focusing on prevention of stricture formation. Stricture formation proceeded until 8 weeks. At that point is stricture treatment point. Our research proceeded proliferation phase of wound healing within 3 weeks. And also similar corrosive burn research papers were referenced.

R 3-8. Line 127: Sentence should be ‘baseline body weight was not statistically significantly different between the control and treatment groups and body weight gain was not statistically significantly different between the two groups.’ – Minor comments

Response: These have been added and revised, as suggested.

R 3-9. The effect of the drug on preventing corrosive strictures seems convincing. It would be know at what point of the three week period the corrosion starts and the drug starts to take effect, if possible.

Response: Thanks for good comment. In future research, we will study according to the wound healing phase point.

R 3-10. Fig 3-A does not indicate what the individual columns represent.

Response: These have been added and revised, as suggested.

R 3-11. Fig 3B- 3E: Statistically significant groups should be indicated using asterisks. For example are the ‘Healthy’ and ‘Treatment’ groups not significantly different in all the histograms? There seems to be a significant difference between the groups in Figures 3D and 3E.

Response: These have been added and revised, as suggested.

R 3-12. Line 192: ‘were prominent lower in the treatment group than those in the control group’ may need to be rephrased to clarify.

Response: These have been revised, as suggested.

R 3-13. Line 32: ‘Formation’ should be ‘formation’ – Minor comments

Response: These have been revised, as suggested.

R 3-14. Section 2.3. Surgical procedure (Line 70): repeat phrase ‘to prevent’ should be removed. – minor comments

Response: These have been revised, as suggested.

R 3-15. Section 2.3. Surgical procedure (Line 85): Sentence should be ‘behavioral and weight changes were monitored weekly.’ – Minor comments

Response: These have been revised, as suggested.

R 3-16. Line 182: ‘Kim et al. reported that blocking the biological action of TGF-β man by an alternative’

Response: These have been revised, as suggested.

R 3-17. Line 206: ‘We surmise that the treatment group experienced (with? Delete word) delayed would healing…’

Response: These have been revised, as suggested.

R 3-18. Line 212: ‘Our study included some limitations’

Response: These have been revised, as suggested.

Reviewer 4 Report

Authors have analyzed the effect of TGF-β Type I Receptor Kinase Inhibitor on the esophageal caustic injury using animal model. 

Authors have clear aims and objectives. Results and discussion are succinct. 

I have few comments:

Please revise the manuscript for grammatical errors. 

What is the rational to have 3 weeks as the end point? Most caustic ingestion, the stricturing effect may be seen until 8 weeks. 

Increase the font size in figure 1. 

Figure 1. the description below the figure states that treatment group "C" has significant narrowing based on values in the images. Is that a mistake. ?

Author Response

Reviewer #4: R 4-1. What is the rational to have 3 weeks as the end point? Most caustic ingestion, the stricturing effect may be seen until 8 weeks.

Response: On this comment, TGF-b1 is a member of the TGF-b superfamily that controls processes such as proliferation. Our research is focusing on prevention of stricture formation. Stricture formation proceeded until 8 weeks. At that point is stricture treatment point. Our research proceeded proliferation phase of wound healing within 3 weeks.

R 4-2. Increase the font size in figure 1.

Response: These have been added and revised, as suggested.

R 4-3. Figure 1. the description below the figure states that treatment group "C" has significant narrowing based on values in the images. Is that a mistake. ?

Response: It is mistake in pptx file. Fig 1-B & Fig 1-C were mis-switch. These have been added and revised.
